# Roles of G4-DNA and G4-RNA in Class Switch Recombination and Additional Regulations in B-Lymphocytes

**DOI:** 10.3390/molecules28031159

**Published:** 2023-01-24

**Authors:** Ophélie Dézé, Brice Laffleur, Michel Cogné

**Affiliations:** Inserm UMR U 1236, University of Rennes, INSERM, EFS Bretagne, 35000 Rennes, France

**Keywords:** G-quadruplex DNA, G4-ligand, class switch recombination, transcriptional regulation, B lymphocytes, G4-RNA

## Abstract

Mature B cells notably diversify immunoglobulin (Ig) production through class switch recombination (CSR), allowing the junction of distant “switch” (S) regions. CSR is initiated by activation-induced deaminase (AID), which targets cytosines adequately exposed within single-stranded DNA of transcribed targeted S regions, with a specific affinity for WRCY motifs. In mammals, G-rich sequences are additionally present in S regions, forming canonical G-quadruplexes (G4s) DNA structures, which favor CSR. Small molecules interacting with G4-DNA (G4 ligands), proved able to regulate CSR in B lymphocytes, either positively (such as for nucleoside diphosphate kinase isoforms) or negatively (such as for RHPS4). G4-DNA is also implicated in the control of transcription, and due to their impact on both CSR and transcriptional regulation, G4-rich sequences likely play a role in the natural history of B cell malignancies. Since G4-DNA stands at multiple locations in the genome, notably within oncogene promoters, it remains to be clarified how it can more specifically promote legitimate CSR in physiology, rather than pathogenic translocation. The specific regulatory role of G4 structures in transcribed DNA and/or in corresponding transcripts and recombination hereby appears as a major issue for understanding immune responses and lymphomagenesis.

## 1. Introduction

A large part of the human genome is composed of repeated sequences. Among them, guanine-rich sequences receive major scientific attention since, instead of the canonical double-helix, self-association of guanines within DNA, RNA (or DNA:RNA hybrids) can locally form a noncanonical four-stranded secondary structure defined as a G-quadruplex (G4). G4 folds after initial pairing of a planar G-tetrad through Hoogsteen hydrogen bonding of four G residues from each strand. This is followed with π–π self-stacking of several proximally located G-tetrads finally stabilized by interactions with cations [1]. According to the number of participating nucleic acid molecules and to the conformation and orientation of the G residues, a G4 can be intramolecular or intermolecular and follow parallel or antiparallel structures, while multimeric assemblies of G4s can also form higher-order structures in the genome [2]. The G4 structure was discovered in 1988 by X-ray diffraction [3] and was then identified throughout the genome of most species, including humans [4]. More importantly, G4s can also be created within the secondary structures of RNA molecules, as identified by reverse transcriptase stalling (rG4-seq) on poly(A)-enriched RNAs [5,6].

G4-DNA has been extensively studied in vitro and is now known to form throughout genomes in vivo, especially in chromosomal telomeres [7] and in regulatory sequences such as promoters and transcriptional enhancers [8]. They are indeed abundant in some gene promoters, notably from oncogenes and from genes involved in the cell response to external stimuli, growth regulation, in cell–cell communication, and in locomotion [9,10,11]. Of importance for the B-cell lineage, G4s are also abundant in Ig variable (V) genes [4], and in the so-called “switch” (S) regions targeted by the process of class switch recombination (CSR) [12] (Figure 1). At such locations, G4s perform regulatory roles but also potentially endanger genome stability by initiating double-strand breaks (DSBs) and translocations. Several well-described experimental techniques can validate the G4-forming capacity of specific sequences, such as nuclear magnetic resonance (NMR) [13], X-ray crystallography [14], circular dichroism spectroscopy [15,16,17], and methods measuring the thermal stability of quadruplexes, namely UV melting [18,19]. However, these biophysical techniques cannot scan the genome of living cells for dynamically identifying the formation of G4s genome-wide in vivo. Thus, several computational (in silico) methods have been developed to detect putative G4s in DNA (and RNA) sequences [20]. G4 structures formed in chromatin in vivo can also be identified directly by antibody-based chromatin immunoprecipitation and high-throughput sequencing (“G4 ChIP-seq”) [8]. From this technique, around 10,000 G4-rich regions were recovered from the genome of a human epidermal keratinocyte cell line (HaCaT), and more than 700,000 G4-DNA sites were identified in the human genome [21]. These data confirmed the predicted enrichment for G4s in regulatory regions such as promoters [8]. 

G4s, whether present in DNA, RNA, or RNA:DNA hybrids [22], participate in multiple genetic regulatory processes, from the positioning of the DNA replication machinery, which is certainly crucial in actively dividing B-cells [23,24], to a role of G4s in RNA translation devoid of any direct effect on gene recombination [25] (Figure 1). Some of the general roles of G4s have already been extensively reviewed elsewhere [26,27,28] and will be briefly mentioned when pertinent to B-cells, while this review will mostly focus on those regulatory aspects which are the most pertinent to the impact of G4s on the control of CSR. This programmed recombination of the Ig-heavy (IgH) chain locus associates with the antigen-induced differentiation of B-cells and with the transient expression of the activation-induced cytidine deaminase (AID). While the specificity of genome remodeling during CSR is instrumental for the development of immune responses and of memory B-cells expressing non-IgM Ig classes, the whole process is also risky for B-cells and can eventually end with genomic aberrations and malignancies. It thus requires careful control, and the role of G4s in IgH genes and transcripts appear to play an important role in this control. 

## 2. General Role of G4s in DNA Accessibility and Gene Regulations, with Pertinence to B-Cells

### 2.1. Unless Unwinded by Helicases, G4s Restrict DNA Accessibility to Various Factors, Notably for Replication and Maintenance of Telomeres 

The conserved TTAGGG DNA repeated motif, which stands at telomeres, forms G4 structures, which maintain the integrity and stability of chromosome ends [29,30]. Telomere sequences can also adopt a two-tetrad G4 conformation in living human cells [31], since the 3′ ends of telomeres are single-stranded [32,33]. Runs of G4s in the telomeric 3′-overhang participate in telomere metabolism and its anchoring in heterochromatin [34]. G4-stabilizing ligands, small drugs interacting with G4s and regulating their function, accordingly lead to telomere shortening [35,36,37]. G4-DNA notably protects telomeres from nuclease and interferes with telomerase, G-rich structures having been shown to inhibit telomerase in vitro [38,39]. Cancer cells use telomerase-dependent (TERT) telomere maintenance or telomerase-independent, i.e., alternative lengthening of telomere (ALT), and stabilization of G4s and R-loops in some transcribed parts of the genome cooperatively enhances ALT-activity [40,41]. ALT could thus constitute a therapeutic target [40]. 

The RecQ helicases WRN and BLM are known to unwind G4s in vitro and to localize to telomeres in vivo where they are required for telomere integrity, providing strong circumstantial evidence for the role of G4 structures at mammalian telomeres [42,43,44,45,46,47]. 

A primary function of RecQ helicases likely is to restore accessibility of G4-DNA to DNA replication, notably at telomeres [48]. The WRN helicase is hence mandatory for preventing telomere loss, and its absence results in chromosomal aberrations such as chromosome fusions [49]. Importantly, the same helicases, and notably BLM, are necessary for optimal CSR and more generally for optimal B-cell differentiation. BLM deficiency can also specifically result in genomic instability in B-cells and in the development of lymphoma [50].

Mutations of other helicases and nucleases unwinding G4s illustrate a broad implication of G4s in DNA replication and are associated with various human diseases, such as in the case of *FANCJ* [51] or *DDX11* [52]. The chromatin remodeling factor ATRX (a SWI/SNF family factor facilitating H3K9 trimethylation in heterochromatin) also carries a helicase domain and associates with the MCM helicase either for unwinding G4-DNA at replication origins or for keeping G4-DNA heterochromatinized and thus preventing G4-induced replication stress [53]. 

Globally and beyond a role at telomeres, R-loops and G4-DNA broadly contribute to the specification of replication origins during the early S phase and origin-proximal G4s act as replication fork barriers [54].

### 2.2. G4-DNA and Regulation of Transcription

G4s can play diverse roles in transcriptional regulation: on the template strand, G4s can directly impede RNA polymerase activity [55,56]. Transcription blockade is due to the formation of unusually stable RNA:DNA hybrids, the stability of which is further exacerbated by triplex formation formed by (GAA)*_n_* repeats, called homopurine-homopyrimidine mirror repeats, under the influence of negative supercoiling [57]. These RNA–DNA hybrids also stimulate transcription termination [58,59,60]. Another possible situation is the formation of G4s upstream of the transcription start site (TSS), which usually inhibit transcription [61] when their formation interferes with the binding of the RNA polymerase II or of transcription factors [62]. G4s can recruit or help the binding of transcription factors such as NF-κB and Sp1 [38], either facilitating or restraining transcription [63]. G4s can promote transcription initiation by recruiting specific transcription factors [64]. 

G4-rich transcription start sites and transcription termination sites were recently reported to be sites for cohesin accumulation and may thereby play a role in the 3D organization of euchromatin [65]. It has also been shown that vimentin, an intermediate filament protein highly expressed within migratory cells, selectively binds to G4 repeats so that soluble vimentin may contribute to the regulation of gene expression and also potentially play a role in the high-order organization of the expressed genome [9].

### 2.3. Links between Transcribed G4s and DNA Breaks in R-Loops 

In some transcribed regions of the genome, transcription of the template DNA strand may readily result in the stabilization of an RNA:DNA hybrid, while the nontemplate strand remains single-stranded within a structure collectively referred as an “R-loop”. An R-loop in a region containing abundant G4-DNA then potentially deserves to be considered as a “G-loop”, with the single-stranded nontemplate strand folded into G4-DNA interspersed with single-stranded DNA (ssDNA) (Figure 2A). G4s and G-loops were reported to favor the occurrence of DNA breaks in G-rich regions flanking oncogenes [66], or at various positions in hypoxic conditions by exposing DNA to base oxidation [67]. Indeed, such breaks can occur during the transcription of the *c-MYC* and *BCL2* genes, two oncogenes that are common *IGH* translocation partners in B-cell lymphomas. Mechanistically, this global trend toward increased DNA breakage could be due to G4-rich regions targeted for DNA deamination by AID and then potentially initiating single-strand breaks and eventually translocations in B lymphomas [66] (see below, Section 6 Illegitimate recombination). 

In mature B lymphocytes, activation and differentiation strongly rely on transcriptional regulation eventually followed by recombination and tightly controlled with regards to the timing of replication, i.e., directly related to the various genetic processes for which we just mentioned the broad contribution of G4s. In this regard, the complex involvement of G4-DNA in the control of CSR deserves thorough analysis and may occur at multiple levels. 

## 3. The Context of CSR and Its Regulation

Once activated, mature B cells can diversify their Ig production through CSR, yielding different classes of antibodies with different constant domains but without affecting antigen binding since CSR preserves expression of the same V exon. Class switching stops the production of IgM and IgD, the only classes expressed by naïve B cells, and instead yields IgG, IgA, or IgE classes with new effector functions, either as membrane-anchored B cell receptors (BCR) in B lymphocytes or as secreted immunoglobulins in plasma cells (Figure 3). The IgH locus contains G4-rich sequences at S regions, which can modulate CSR, and therefore represent potential therapeutic targets (see Section 8). While bringing new functions to switched B cells and switched antibodies, class switching does not affect the antigen specificity, since it only reorganizes the constant gene cluster of the Ig heavy locus. This recombination process hence associates the most upstream IgH exon, encoding the VDJ domain, with a new downstream constant Ig gene.

### 3.1. The CSR Machinery

Human and mouse IgH constant genes (C_H_) are all preceded by S sequences (except for Cδ), themselves preceded by germline cytokine-dependent promoters (the so-called “I” promoters). The IgH locus undergoes various gene remodeling events in activated mature B cells and notably within the germinal center (GC) and, to a lower extent, during extra-follicular B cell activation. Activated GC B cells notably feature high expression of AID, which is the enzyme initiating somatic hypermutation and CSR [68]. However, AID by itself is not sufficient for CSR to occur, and a large part of CSR regulation involves regulated accessibility of specific targeted regions within the IgH locus. A minimal level of CSR is even detectable in an accessible IgH locus in the absence of AID, highlighting the importance of the structure and the accessibility of S regions to recombination [69]. These data show that transcribed S regions undergo rare DNA breaks even in the absence of AID, likely due to the presence of G4s and R-loops at these regions. Regulated accessibility to CSR is notably (but not only) related to transcriptional regulation of the S regions targeted for CSR [70,71,72,73]. These S regions have a unique position upstream of C_H_ genes, where the preceding “I” promoter can yield transcription in adequately activated B cells before any CSR event (i.e., in a germline configuration). The IgH locus C_H_ cluster thus includes a series of consecutive “I_H_-S_H_-C_H_” germline transcription units [74,75]. When DNA breaks simultaneously affect the S regions of the upstream I_µ_-S_µ_-C_µ_ (donor) unit and the downstream I_X_-S_X_-C_X_ (acceptor) unit, DNA ligation can occur between both distant broken S regions by the DNA repair machinery of nonhomologous end-joining (NHEJ). Such an S–S junction deletes the intervening genes and features CSR. While I_H_-S_H_-C_H_ germline transcripts are sterile, their specific organization with the S region standing within an intron (downstream of the noncoding I exon) is mandatory for CSR to occur, as will be discussed in detail below [72,76]. 

### 3.2. S Region Structures 

In addition to their specific intronic position, the S regions preceding the Ig C_H_ genes possess unique characteristics. This notably consists in the presence of highly repetitive DNA with a specific nucleotide composition, which participates in their specific targeting by AID for efficient CSR. To be functional, S regions need to be made up of such repetitive DNA, but both the sequence and the number of tandem repeats vary between the various S regions and between species, notably between mice and humans, as reviewed in Dunnick et al. [77]. Mouse and human S regions are respectively composed of 1–10 kilobases (kb) [12,77] and 1–12 kb [78] of tandem repeats, with a G-rich nontemplate strand DNA. 

When S regions are transcribed, a supercoiling relaxation occurs, and the template strand can readily form R-loops [79]. The addition of exogenous RNA corresponding to the Sα sequence (as transfected *transacting* factors)*,* in the absence of transcription in cis by RNA polymerase, did not result in the creation of R-loops at the untranscribed Sα region, demonstrating that the formation of R-loops does not rely on transcripts but on transcription per se. The formation of such RNA:DNA hybrids in transcriptionally active S regions is critical for CSR [79]. Indeed, the deletion of the 10 kb of the S_γ1_ region, eliminating the 8 kb of conserved S_γ1_ repeats, almost abrogates IgG1 CSR [80]. 

### 3.3. Transcriptional Regulation of CSR and the Role of the IgH 3′RR 

CSR relies on the presence of an IgH locus 3′ regulatory region (3′RR), which governs transcription within a topologically associated domain (TAD) and lies in-between the Cα gene and the 3′ boundary of the IgH TAD, itself marked by CTCF-binding elements (CBEs). The 3′RR (which is duplicated in humans) assembles several enhancers according to a palindromic architecture, which favors functional synergies and altogether constitutes a super-enhancer [81,82,83,84]. 3′RR enhancers are separated by repetitive DNA that resemble S regions (like-switch, LS regions) and are eventually G4-rich in either the template strand or the nontemplate DNA strand [83,85,86] (Figure 2B)**,** which contributes to the 3′RR structure and its targeting by AID. Before the onset of CSR, the 3′RR promotes germline I_H_-S_H_-C_H_ transcription of constant genes targeted for CSR, then contributing to their accessibility to recombination and helping to generate ssDNA on the nontemplate strand of the S region [74,75,85,87,88]. During CSR, the 3D architecture of the locus is remodeled, and the 3′RR and its 3′ flanking CBEs form a large loop with the Eµ enhancer [89], within which S regions align via loop extrusion [90]. Recent studies revealed a crucial role of cohesin in anchoring loop extrusion upstream of CSR [90], with the CBEs downstream of the 3′RR as an anchor [91]. Such a loop forms a stable synapse between S regions, in parallel transcribed and targeted by AID for initiating DSBs. This synapse facilitates deletional CSR through the ligation of DSBs from donor and acceptor S regions [77]. The process thus appends the upstream VDJ region to a new downstream C_H_ gene and switches activated B cells from the expression of IgM and IgD to IgG, IgE, or IgA (Figure 3). In rare cases, DNA breaks standing immediately upstream of Cδ, despite the absence of any classical S region and G4s at this position, can also restrict Ig production to IgD only [71,92,93]. Aside from this classical loop extrusion model with cis-recombination, it is also noticeable that CSR can occur in trans and join S regions from both IgH alleles [94,95,96].

Positive and negative regulatory elements act alternatively to control the accessibility of the IgH locus and to ensure transcriptional regulation of CSR. The multiple promoters of the locus interact with several enhancer, silencer, and insulating elements. The 3′RR encompasses a large piece of noncoding DNA, including a cluster of dispersed DNase hypersensitive sites (hs) bound by transcriptional factors and corresponding to the core enhancers [81,85,97]. 3′RR enhancers fall into two distinct modules, acting in a relay race to ensure fine-tuned BCR expression in naïve B cells (at steps dependent on Eµ) and antigen-dependent locus remodeling in mature stages (then with the 3′RR as the master control element) [98]. Beyond transcription in activated cells, the 3′RR role is mandatory for somatic hypermutation of V regions and CSR, i.e., for all IgH remodeling events relying on AID [74,87,88,99,100]. In the hierarchy of enhancer element strength, the 3′RR also exerts some control on the transcription and the function of Eµ [101]. 

### 3.4. Structure and Role of Germline Transcripts

As an early event following B cell activation, transcription by RNA polymerase II initiates at germline promoters upstream of I exons and continues through S regions, i.e., within S introns [102,103]. Noncoding transcription also occurs in parallel within the 3′RR itself, generating enhancer-associated RNAs (eRNAs) [86]. Each C_H_ region (except Cδ) includes an individual transcriptional unit that can produce noncoding germline transcripts (GLTs) upon adequate stimulation, for example, IFNγ for Cγ2a GLTs, IL4 for Cε GLTs, TGFβ for Cα GLTs, etc. [71,104,105]. Transcription produces a primary GLT through the intervening I-exon, intronic S region, and C_H_ exon. Splicing creates a processed GLT and an intronic S region lariat that can undergo debranching to become a linear S region transcript [106]. GLTs transcription depends on several regulatory elements of the IgH locus, and their transcription and splicing are mandatory for CSR.

Radbruch et al. showed that in mice lacking the I_γ1_ exon donor splice site (DSS), no stable GLTs were detected, suggesting that the absence of a DSS prevented the stable transcription of the downstream S_γ1_ region and the C_γ1_ exons. In these B cells, the deletion of the I_γ1-_exon DSS reduced RNA polymerase II pausing and active chromatin marks in the S_γ1_ region, essential elements for the opening of the chromatin in targeted S regions, and resulted in a dramatic loss of CSR [73,107]. A recent study confirmed these findings by using antisense oligonucleotides (ASOs) which targeted either the Iµ DSS upstream of the Sµ donor or the I_γ1_ DSS upstream of the S_γ1_ acceptor region, hereby inhibiting both splicing and CSR at these positions [108]. Thus, CSR is dependent on accurate splicing, and either I-exons or I_γ_ exon DSS recognition is necessary for the regulation of CSR. Further investigations in both mouse and human B cells will be necessary to check whether masking the Iµ exon DSS would decrease CSR to all isotypes. Conflicting results were indeed reported in mice lacking the constitutive Iµ DSS but with normal serum Ig levels, while B cells then produced alternative “Iµ-like” GLTs, which likely preserved a contribution to CSR [109]. 

Transcription of the S regions promotes the occurrence of R-loops at positions where RNA polymerase II is stalled, while stable RNA:DNA heteroduplex associations are formed by the RNA transcript and the template DNA strand, then increasing the accessibility for AID of the displaced single-stranded nontemplate strand [110]. 

To better understand the role of GLTs, several researchers modified the Cµ [111,112], C_γ1_ [72,73,107], C_γ2_ [76,113], C_ε_ [104,105], and C_α_ [114] transcriptional control elements. These studies suggested that transcription of C_H_ loci, and more precisely the generation of stable S transcripts, was necessary for efficient CSR. Moreover, studies of the I.29 mouse B cell lymphoma, which undergoes CSR from membrane-bound IgM to IgA, allowed us to characterize transcripts from the unrearranged Cα region in IgM ^+^ I.29 cells [102] and to show that the magnitude of Cα GLTs correlates with the efficiency of IgA CSR [103].

### 3.5. Post-Transcriptional Regulation of GLTs and Their Processing by the RNA Exosome

Transcription can create RNA:DNA hybrids at multiple locations in the genome. At S regions, these RNA:DNA hybrids limit DNA accessibility to AID on the template strand. AID preferentially targets cytosines for deamination [115,116] on the accessible nontemplate ssDNA. It has been demonstrated that AID interacts with the RNA exosome, which degrades these noncoding GLTs at S regions [117], thus generating two ssDNA and allowing accessibility to the template strand for AID. Dedicated mouse models, specifically deleting either the *Exosc3*, *Exosc10*, or *Dis3* gene, confirmed this discovery [118,119,120]. The RNA exosome is the predominant 3′ exoribonuclease in mammalian cells and is responsible for the degradation and/or 3′ end processing of a variety of noncoding RNAs [121]. This complex is composed of nine core and two catalytic subunits, and it participates in the resolution of R-loops and hereby contributes to optimal CSR. 

Persistence of hybridized GLTs within R-loops in the absence of the RNA exosome also perturbs the progression of the cohesin complex during the process of loop extrusion preceding the CSR, resulting in altered IgH synapsis and decreased CSR but increased translocations [119]. This defect in cohesin scanning could also be a consequence of G4s accumulation within the long R-loops of the S regions, as G4s were recently reported to be sites for cohesin accumulation at G4-rich regions [65]. The secondary structure of G4s likely slows down or stops the cohesin progression and hereby probably contributes to genome organization. Finally, it has been shown that GLTs stability is dependent on the epitranscriptomic mark N^6^-methyladenosine (m6A), which is necessary to address the Sµ GLTs for degradation by the RNA exosome complex [122]. 

Globally, GLTs are necessary for CSR; they must be produced, modified, and finally degraded by the RNA surveillance machinery and the RNA exosome complex for efficient CSR and to prevent aberrant translocations. 

## 4. Connection between G4s and CSR 

### 4.1. Role of G4-DNA at Transcribed S Regions and in the Resolution of G4s/R-Loops Conflicts

Although AID can target AT-rich S regions from amphibians during CSR, G4-DNA is well known to abundantly form on the nontemplate strand of transcribed mammalian S regions [1]. In silico analysis (for example using the G4s-hunter algorithm, either http://bioinformatics.cruk.cam.ac.uk/G4Hunter or https://www.g4-society.org/online-tools, both accessed on 21 Octobre 2022) notably shows that G4-DNA is abundant at mouse and human IgH S regions [123] (Figure 2B). There are also multiple functional indications that G4s are implicated in CSR regulation and promote gene recombination [124]. G4-DNA notably promotes the occurrence of DNA breaks [125]. Among the seminal studies of the molecular properties of S sequences, Wells and colleagues cloned various G-rich S_α_ repeats into plasmids in a search of peculiar DNA structures [126] and showed that S_α_ repeats adopted a non-B DNA structure, which is characterized by supercoil-dependent endonuclease cleavage and sensitivity to chemical probes, suggesting a potential intramolecular triple-strand. Moreover, Sen and Gilbert showed by electrophoretic mobility shift assays that S regions adopt the canonical model of a parallel, four-stranded G4-DNA structure diffraction [3]. 

R-loops and G4s can act as physical impediments to DNA and RNA polymerases during replication and transcription, where G4s stabilize R-loop structures and facilitate the local recruitment and oligomerization of AID [127]. However, the detailed molecular contribution of G4s to CSR remains unclear, and while their presence favors CSR, their stabilization by G4s-ligands by contrast impedes CSR [123], suggesting that the contribution of G4-DNA follows a dynamic scheme. Whether the abovementioned fragility of G4-DNA in hypoxic conditions might contribute to the process of CSR is currently unclear. While B cell activation largely happens in vivo in hypoxic lymphoid structures such as GCs, conflicting data have been published about the connection between hypoxia and class switching, mentioning both increased CSR breaks in B cells cultured in hypoxia conditions in vitro and increased CSR to the Cα gene in vivo but also decreased AID expression in vivo and decreased CSR to IgG2 in a mouse model experimentally exposing GCs to hypoxia [128,129]. These ambiguous global effects of hypoxia might be obscured by the fact that AID expression is lowered by hypoxia, which may then counterbalance a simultaneous increase in DNA breaks in G4-rich S regions [129].

As previously mentioned, CSR occurs in transcribed G-rich S regions, forming RNA:DNA hybrids on the template strand and exposing single-stranded R-loops on the nontemplate strand, which is then a substrate for AID [71,122]. This likely contributes to the prevalence of orientation-dependent CSR, joining distant breaks both initiated on the nontemplate strand [80] (although orientation-independent CSR has also been experimentally reported after breaks involving short palindromic sequences instead of G-rich sequences [130]) (Figure 3B). 

A pair of nucleoside diphosphate kinase (NME) isoforms, one of them binding G4s, are novel players in the CSR process. They were identified (using a reverse ChIP proteomic screen and a gel shift with single-stranded DNA) by searching proteins associated with CSR DSBs in B cell lines and mouse primary B cells. NME1 binds S regions before the formation of G-loops and represses the initiation of CSR. When G-loops are formed upon stimulation, NME1 then dissociates from activated S regions, whereas NME2 binds G-loops and promotes CSR [131] (Figure 3A,C). The NME1/NME2 pair thus coordinately modulate G-loop accessibility and CSR. 

In addition to the role of G4s at a DNA level, there is abundant direct and indirect evidence that G4s and/or equivalent structures forming in parallel on RNA transcripts from these regions are implicated in CSR regulation. It remains unclear to what extent AID directly targets DNA and/or requires G4-RNA intermediates, and this remains a controversial topic. 

### 4.2. Role of G4-RNA Structures within S Region Transcripts 

In the context of mammalian S regions, the G-rich nontemplate single strand DNA and the corresponding nascent RNA can both form G4s motifs, the latter being called G4-RNA. G4-RNAs can be found in more than 3,000 human mRNAs [5,6]. Transcriptomic profiling of G4-RNAs is possible via G4-RNAs-specific precipitation (G4 RP) using the G4s-specific probe, BioTASQ [132]. 

For the specific situation of S region GLTs, several studies showed that AID can directly bind S transcripts through G4-RNAs. As mentioned above, GLTs undergo splicing and are then liberated as processed GLTs, while the S region lariat remains annealed as part of the R-loop. Debranching and folding of the lariat into G4 secondary RNA structures likely contribute to the recruitment of AID via AID–RNA binding. Yewdell and Chaudhuri proposed models for RNA-dependent targeting of AID during CSR [106]. They notably postulated a role of AID–RNA complexes in trans. In this situation, the S region lariat is debranched to form a linear S region transcript, which can fold into a G4 secondary RNA structure. Then, either this structure is bound by AID, and the following complex then binds on the complementary DNA strand, or the RNA first binds the complementary DNA strand, and both are then bound by AID. In another model proposing the targeting of AID–RNA complexes in cis, processed nascent GLTs would remain attached to the template DNA strand at the position of R-loops. 

Whatever the model, AID binds structured substrates G4-DNA [124] and efficiently yields mutation and DSBs clusters at the positions of S regions featuring ‘‘G-loops’’, and this may also help to recruit CSR cofactors [133]. The situation of the IgH locus is worth comparison with other contexts, where R-loops in viral RNA (from HIV, Zika, Hepatitis B, SV40, etc.) topologically control its adenosine methylation and thus show colocalization of G4-RNAs with the epitranscriptomic mark m6A [134]. Such a role remains to be explored in B cells, where it could potentially interfere with m6A-dependent processing of GLTs by the RNA exosome [122]. The m6A mark allows RNA exosome binding for degradation of RNA:DNA hybrids, and it was recently shown that m6A modifications controlled not only G4-RNAs but also G4-DNAs formation, then regulating the biological functions of these structures [135]. Of note, RNA sequence also influences RNA binding to lipid membranes. This interaction is increased by G4-RNAs [136], and this may participate into the functional role of G4-rich RNA during biological processes by tethering some G4-RNAs. 

It is noticeable that a mutation in the putative RNA-binding domain of AID impairs its recruitment to S regions, inhibiting CSR similarly to the inhibition of RNA processing [137]. Inhibition of CSR was also obtained by inhibiting a specific step of the processing of S introns: the debranching of the lariat by the DBR1 enzyme [137]. Expression of switch RNA in trans then rescued the CSR defect in DBR1-deficient B cells [137]. Availability of debranched RNA copies of S regions may thus contribute to the subsequent generation of G4-RNAs that participate in guiding AID to specific DNA S regions through RNA:DNA base pairing as a “collaboration” between G4-RNAs and G4-DNAs. In addition, by focusing on G4s present in intronic S region RNA, Ribeiro de Almeida et al. showed that the RNA helicase DDX1 unwinds G4-RNAs structures, allowing these RNAs to participate to R-loops in vitro and in vivo. Therefore, in this model, R-loops at S regions are formed post-transcriptionally in trans and are dependent on DDX1 and G4-RNAs. Stabilizing G4-RNAs with G4 ligands such as pyridostatin or inducing the expression of DDX1 ATPase-deficient mutant accordingly reduces CSR [138]. Moreover, alternative lariat sequences could avoid the fixation of DDX1 to G4-RNAs that participate in guiding AID to specific S regions through RNA:DNA base pairing. CSR would then rely on connections between G4-RNAs on S region transcripts and DNA at R-loops [138]. 

### 4.3. Connections between CSR and DNA Replication 

In addition to its connections with transcription, CSR is temporally and physically connected with the progression of DNA replication through the IgH locus and AID-dependent DNA breaks occur and are mostly repaired within the G1 phase [139]. While R-loops contribute as mentioned above to the specification of replication origins, CSR efficiency notably depends on and correlates with the activity of these origins in S regions, and G4-DNA participates in CSR regulation in an indirect way in mouse B cell lines and in primary splenic B cells [140]. DNA replication across S regions also regulates CSR in an R-loop-dependent manner. Wiedemann et al. demonstrated that the origin of replication is independent of AID and of DNA breaks but indeed mostly relies on G4-DNA, so that facultative replication origins assemble at R-loops and contribute to the synapsis of S regions targeted by CSR [140]. Actually, at the IgH locus, as in other parts of the genome, G4-DNA impacts the binding of the origin recognition complex (ORC) and requires the replicative helicase activity of MCM [141]. In the G1 phase, IgH transcription allows R-loop formation including G4-DNAs and then triggers the activation of facultative G4-rich replication origins within S regions [140]. Since replication origins located within the same TAD tend to be physically clustered, this may favor the synapsis of S regions including such origins [140]. In this way, CSR would not only be coordinated with cell proliferation but also physically facilitated by the mechanistic aspects of DNA replication during the G1-phase (Figure 3C). 

### 4.4. A Role of G4-DNA in IgH Locus High-Dimensional Organization

As mentioned above, the physiology of IgH locus expression and recombination is based on programmed changes of the IgH locus 3D organization, based on long-range interactions between promoters, enhancers, and regions targeted for recombination. These dynamic changes are notably interpreted through the loop extrusion model, which allows the synapsing of distant recombination sites prior to V(D)J recombination and class-switching [90,142]. The role of G4-DNA in these events is not currently demonstrated, but it is striking to note that G4-DNA is abundantly mapped at the position of regulatory chromatin and may then play a role in the organization of TADs [8]. Presence of the G4-DNA within the IgH 3′RR might then play a role in the organization of the IgH TAD. Of note, the architectural factor YY1, known both to bind the 3′RR and play a role in DNA looping, is another known binder of G4-DNA [143]. HP1α also binds G4-DNA and is known for its role in the organization of separate domains of heterochromatin or of transcriptionally active euchromatin [34]. The abovementioned superimposition of G4-DNA [65], or G-rich R-loops [119] with regions bound by cohesin, and by soluble vimentin [9] further argues for an architectural role of G4-DNA. All these elements are likely to be crucial for the process of CSR, which strongly relies on 3D interactions between germline promoters, the 3′RR, and the targeted S regions within an active IgH TAD [89]. 

## 5. A Role of G4s in the Regulation of Locus Suicide Recombination (LSR) 

As mentioned above, CSR is controlled in both mice and humans by the 3′RR super-enhancer located downstream of the Cα genes (i.e., with two copies in the human locus, the 3′RR1 downstream of Cα1 and the 3′RR2 downstream of Cα2) [97]. As in mice and in all mammals where IgH sequences are available, the human 3′RR enhancers are embedded within a large stretch of repetitive DNA with inverted repeats, direct repeats, and LS regions resembling S regions [81,83,85,144,145,146]. While AID-dependent CSR generates DNA breaks between two S regions, the fate of B cells can also be altered more dramatically by another AID-dependent process similar to CSR but joining DNA breaks from a S region with another break from the 3′RR, then featuring “locus suicide recombination” (LSR). Unlike CSR, which excises part of the IgH C_H_ gene cluster, LSR results in the deletion of all constant genes by joining DNA breaks between Sμ and the 3’RR, thereby abrogating BCR expression (or precluding Ig secretion for cells engaged in plasma cell differentiation). This atypical CSR-like event first observed in mouse B cells [86] also abundantly occurs in human B cells [146]. Sette et al. have highlighted G4 structures in vitro outside S regions and notably close to the 3′RR hs1.2 enhancer [38], as shown by analysis based on the G4-Hunter (Figure 2B). We currently have little information regarding the LSR regulation, the study of which is more complex that for CSR since BCR loss after LSR rapidly results into B cell death and makes the detection of LSR junctions about 100-fold less efficient than for CSR junctions [147]. By analogy to CSR, contribution of G4-DNAs and/or G4 RNAs to the regulation of LSR is quite possible. A missing part of the CSR/LSR puzzle currently lies in understanding what elements could preferentially target AID S regions or LS regions of the 3’RR, and G4-DNA and R-loops might contribute to such a regulation. 

## 6. G4s and Illegitimate Recombination 

Aside from physiology, the involvement of G4s in pathological processes has been reported by multiple studies. Xu et al. showed by chromatin immunoprecipitation sequencing (ChIP-seq) that AID hotspots were highly enriched for G4 structures in activated B cells and in lymphoma cells in vitro. In some B cell malignancies, mutations and/or gene amplification of the *BCL2* and *MYC* oncogenes participate to cell transformation. Ninety-seven percent of the *BCL2* mutations occur at G4-rich positions, which overlap those of AID binding [148]. Therefore, G4 targeting by illegitimate mutations or recombination participates in the loss of genomic integrity, a critical step in B lymphomagenesis [148]. Due to their role both in the transcription of some oncogenes and in the process of illegitimate recombination, G4s could thus also participate in genomic instability during the progression of malignancies. Likewise, the in vitro transcription of the *c-MYC* and *BCL6* genes has been reported to promote R-loop formation [66,149]. As mentioned before, G4 structures formed within the ssDNA portion of the R-loop are called G-loops, and AID was observed to bind G-loop structures in vitro by using electron microscopy [149]. This suggests that the G4s binding property of AID may contribute to aberrant targeting and oncogenic IgH translocations. 

## 7. Mutations of Various Nuclear Factors with G4-Dependence

While G4s promote the recruitment and oligomerization of AID, which initiates CSR (Figure 3C), some mutations of the AID sequence can accordingly reduce CSR in a G4-dependent manner. Yewdell et al. indeed generated through CRISPR-Cas9 targeted mutagenesis a mouse strain with the same G133V mutation observed in patients with hyper-IgM syndrome [150,151,152]. This G133V variant remains catalytically active but is altered in its ability to bind G4s [150] (Figure 3E). The mutation hereby decreases CSR and IgA, IgG1, IgG2b/c, and IgG3secretion drastically, while increasing unswitched IgM secretion. AID^G133V^ has genome-wide chromatin localization defects especially in the Sµ region [150]. Of note, while CSR is principally AID-dependent (and not only G4-dependent), a recent study showed that the structure of S regions and their programmed accessibility are sufficient for preserving a basal level of CSR in AID-independent conditions [69]. Indeed, in both AID-deficient mice and AID-mutant patients, CSR junctions remain detectable at low levels upon B cell activation. These AID-independent CSR events likely rely on the presence of R-loops and G4s initiating DSBs by themselves, explaining why DSBs remain focused on S regions in such conditions.

Some other enzymatic deficiencies perturb mature B cells homeostasis and are associated with accumulation of G4s and R-loop structures [153]. Enzymes of the TET family and especially TET2 and TET3 regulate enhancer activity and DNA methylation dynamics during B cell development [154,155], while *TET2* mutations or defects are frequent in hematological malignancies such as diffuse large B cell lymphoma [156,157]. R-loops and G4-DNA (as detected by a G4-specific antibody or with a fluorescent G4-ligand) accumulate in TET2 and TET3-deficient B cells, together with abundant DNA DSBs in IgH S regions and with a genome-wide increase in translocations implicating the IgH locus [153]. This suggests that G4s and R-loops could be therapeutic targets in cancers with TET loss-of-function. 

## 8. Pharmacological G4 Targeting and Potential Implications for B Cells and CSR 

Many of the regulations depending on G4s can also be pharmacologically modulated by G4 ligands, i.e., small molecules stabilizing these structures. In living cells, most G4 ligands have widespread effects, and they have been shown to affect cell growth through multiple mechanisms, by altering telomere stability, replication, transcription, RNA metabolism, and mitochondrial maintenance [158]. Since G4 ligands such as pyridostatin increase the stability of telomeric G4s, they notably have antiproliferative activity, because the G4 structures cannot be extended by telomerase, an enzyme overexpressed in many actively proliferating cells, notably cancer cells [41,159] (Figure 3). Re-expression of the telomerase can restore telomere functions, consistent with long-term cancer cell proliferation [160]. Pyridostatin induces dysfunctional telomeres in cancer cells with the uncapping of POT1, essential for the replication of chromosome termini, resulting in DNA damage signaling activation and notably showing antitumoral activity against BRCA1/2-deficient tumors [161]. The less potent G4-ligand, RR110, lacks this effect [41]. Targeting G4s at oncogene promoters could also be of interest for cancer therapy, and G4 ligands binding the *c-MYC* promoter or telomeric G4s were shown to downregulate both the *c-MYC* and *hTERT* gene expression, while upregulating γ-H2AX and 53BP1 due to DNA damage, yielding in vivo antitumor effect in tumor-engrafted mice [162]. 

Many studies are currently focusing on modular G4-ligands that interact with the loops and grooves of G4-DNAs in cancer cells. This leads to more selective compounds aiming to act as specific anticancer chemotherapeutic agents [162,163]. In addition, positive feedback from G4-stabilizing agents is possible. BRCA1 and BRCA2, two tumor suppressor genes that interact with G4 structures [164], participate in homologous recombination and thus allow the repair of DNA DSBs. When these proteins are mutated, in breast cancer, the G4 stabilizer CX-5461 can block replication forks and result in unrepaired DNA breaks. CX-5461 was also recently shown to decrease cell viability in ATRX-deficient glioma and BRCA1/BRCA2-deficient tumor cells [165,166]. Since the BRCA and NHEJ signaling pathways are required for the repair of DNA damage induced by CX-5461, defective repair results in cell lethality [166], which pushed CX-5461 in current phase I/II cancer clinical trials [167]. It is also possible to control G4 dissociation by the ionic environment, concentration, and temperature, with G4 lifetimes being longer in KCl than in NaCl and LiCl [168], but this has not yet been modulated in cancer cells. 

The role of G4s in cancer could involve multiple proteins that process or bind G4s, AID being one of them. This essential enzyme of the adaptative immune system, which normally diversifies antibody generation, indeed also participates to oncogenic events, its aberrant expression being a key driver of lymphoid cancers [169,170,171]. Actually, a clear link exists between AID and the translocation of *c-MYC* in Burkitt lymphoma, and the mechanistic evidence has been shown in vitro by Duquette et al. [149]. MYC mandates tumor cell fate and orchestrates changes in the tumor microenvironment, including the activation of angiogenesis and suppression of the host immune response. Using electron microscopy, it has been demonstrated that AID binds G-loops, so G4s form during transcription of the *c-MYC* gene as well as IgH S region. These G-loops were mapped and overlapped to the breakpoints related to *c-MYC* translocation.

In nonmalignant B cells, some G4 ligands were shown to also affect CSR [123] (Figure 3C,E). Since inappropriate humoral immune responses involving proinflammatory class-switched Ig can lead to immunopathology, such drugs able to directly and specifically modulate CSR might be of strong therapeutic interest. In vitro, in primary B cells and in vivo in immunized mice, treatments with RHPS4 (3,11-difluoro-6,8,13-trimethyl-8H-quino [4,3,2-kl]acridinium methosulfate), a G4-stabilizing agent, decreases CSR. By quantifying the GLTs specific for the pre-CSR stage (Iµ-Cµ and Iγ-Cγ) and the post-CSR transcripts (Iµ-Cγ), post-CSR transcripts decreased, while the pre-CSR increased, and Sµ–Sγ junctions decreased by four-fold with RHPS4 [123]. This drug was indeed shown to decrease AID binding to S regions, hereby decreasing CSR and subsequent secretion of class-switched Ig. In vitro, RHPS4 decreased CSR in stimulated B cells without major side effects on cell growth, while in an in vivo mouse model it also reduced the development of airway inflammation, suggesting that G4 ligands might have a therapeutic interest in autoimmune or allergic conditions, which notably involve class-switched antibodies [123]. However, given the multiple roles of G4s in gene regulation, their nonspecific targeting throughout in the genome is clearly not satisfactory. The current development of new strategies able to target G4 structures within a given sequence [162,172] would certainly open new opportunities by providing therapeutic means to target a single S region and then inhibit CSR to a single Ig gene with minimal side effects.

Bossaert et al. also showed that transcription-associated topoisomerase 2α (TOP2A) activity is a major effector of the cytotoxicity induced by the clastogenic G4 ligands pyridostatin and CX-5461 in cell lines. Using an unbiased genetic approach, it was shown that these G4 ligands prevent RNA polymerase II elongation and promote the DNA cleavage by TOP2A while inhibiting repair [173]. Pyridostatin thus acts synergistically with inhibitors of DNA repair [174,175]. In human cells, upon stabilization by pyridostatin, G4 structures interact with the NELF complex, which modulates the cellular response to G4 ligands [176]. While TOP2A is almost exclusively expressed in proliferating cells and is needed for DNA replication, sister chromatid segregation, and transcription, TOP2B is expressed throughout the cell cycle and releases torsional stress at sites of transcription [177,178] (as TOP2A might eventually also do). Bruno et al. previously showed a major role for TOP2A in the induction of DNA DSBs upon CX-5461 treatment [179]. Since pyridostatin and CX-5461 inhibit the re-ligation of DNA DSBs, they would be expected to decrease the repair of S–S junctions and therefore constitute potential pharmacological modulators of CSR. NELF could also regulate CSR in a G4-ligand dependent manner [176]. As for CX-5461, which is currently in phase I/II clinical trials for cancer treatments [167], other G4-ligands are potential anticancer agents. Of note, the anticancer drug etoposide, which traps TOP2 in its DNA cleaving form and thereby prevents ligation, is used for treating myeloid leukemia.

Pharmacological targeting of the nuclear factors associated with the folding or the processing of G4s might indeed be of therapeutic interest. Notably, the TOP1 topoisomerase also participates to CSR regulation by limiting cotranscriptional G4s formation, and its inhibition by the specific inhibitor camptothecin can inhibit CSR, while its genetic knock-down by contrast enhances CSR [173]. Interestingly, AID itself mediates such a “physiological knock-down” of TOP1 by editing a miRNA, which lowers TOP1 translation [170].

## 9. Conclusions

While telomeres, with their long terminal repeats, are the largest G4 reservoir in the genome, G4s have multiple functions in living cells beyond these sequences. G4s notably have major roles in the B cell lineage not only related to genomic instability in lymphoproliferative disease but also to the physiology of CSR, as detailed along the lines of this review. CSR can be regulated by G4 structures in various and eventually opposite ways. On the one hand, G4 motifs promote DNA accessibility to AID and favor the occurrence of DSBs during the humoral immune response. On the other hand, G4-DNA bound by various natural or pharmacological ligands can inhibit CSR. The role of G4s in S regions and S transcripts globally remains incompletely understood and deserves to be further explored both for understanding physiology and because G4s, R-loops and G-loops may constitute useful therapeutic targets.

Many G4-dependent cellular processes related to cancer are indeed considered “druggable” using molecules such as RHPS4, pyridostatin, and CX-5491. CSR could thus also be considered as a “druggable” process. Actually, drugs such as pyridostatin promote DNA breaks but prevent their re-ligation. A G4-stabilizing agent such as RHPS4 decreases both CSR and decreases Ig secretion. Such effects might be of interest in autoimmune or immuno-allergic conditions. They may also be pertinent to tumor immunology, given the roles of B cells, which infiltrate solid tumors, either promoting antitumor immunity or, by contrast, producing immune complexes with tumor antigens, which bind M2 macrophages and lead to tumor-promoting deleterious inflammation [180,181,182].

In all these contexts, pharmacological control of CSR might be of interest for decreasing the production of the most proinflammatory class-switched Ig and rather favor the production of IgM. However, the most attractive control of humoral responses would reside in the ability to precisely modulate the secretion of a single Ig class for either promoting or tempering inflammation, which may rely on the combination of G4-ligands with other immunomodulatory strategies. By analogy to the strategies pursued toward the use of specific G4-ligands for cancer therapy and along the ongoing refinement of sequence-specific ligands [162,163], the design of Ig class-specific agents able to act on the accessibility of a given S region and not on the global CSR process should also be a therapeutic grail.

## Figures and Tables

**Figure 1 molecules-28-01159-f001:**
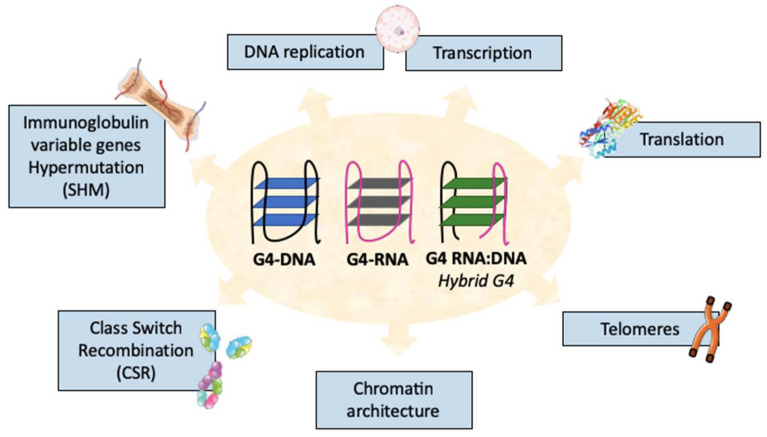
Multiple biological functions of G-quadruplexes (G4s). G4s have different conformations such as G4-DNA, G4-RNA, or G4-RNA:DNA called hybrid G4s. These structures are found at many locations in the genome and are implicated in multiple biological processes.

**Figure 2 molecules-28-01159-f002:**
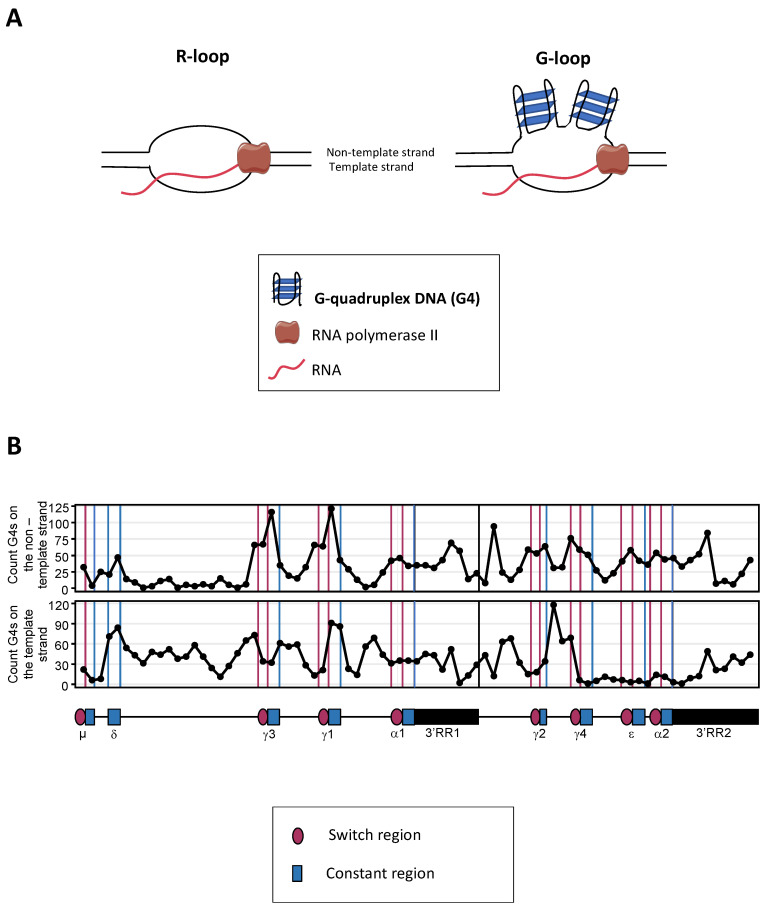
G4s at S regions. (**A**) Schematic representation of R-loops and G-loops. Transcription can create RNA:DNA hybrids, also called R-loops (left). When the nontemplate DNA strand of R-loops is G-rich, it can create G4s, and these structures are then called G-loops (right); (**B**) Representation of G4s located in the human IgH locus constant gene cluster (from Sµ to the end of the 3′RR2) on the coding strand (top) and the template strand (bottom). This representation has been made by processing the IgH sequence with the G4-Hunter algorithm (https://www.g4-society.org/online-tools, accessed on 21 October 2022).

**Figure 3 molecules-28-01159-f003:**
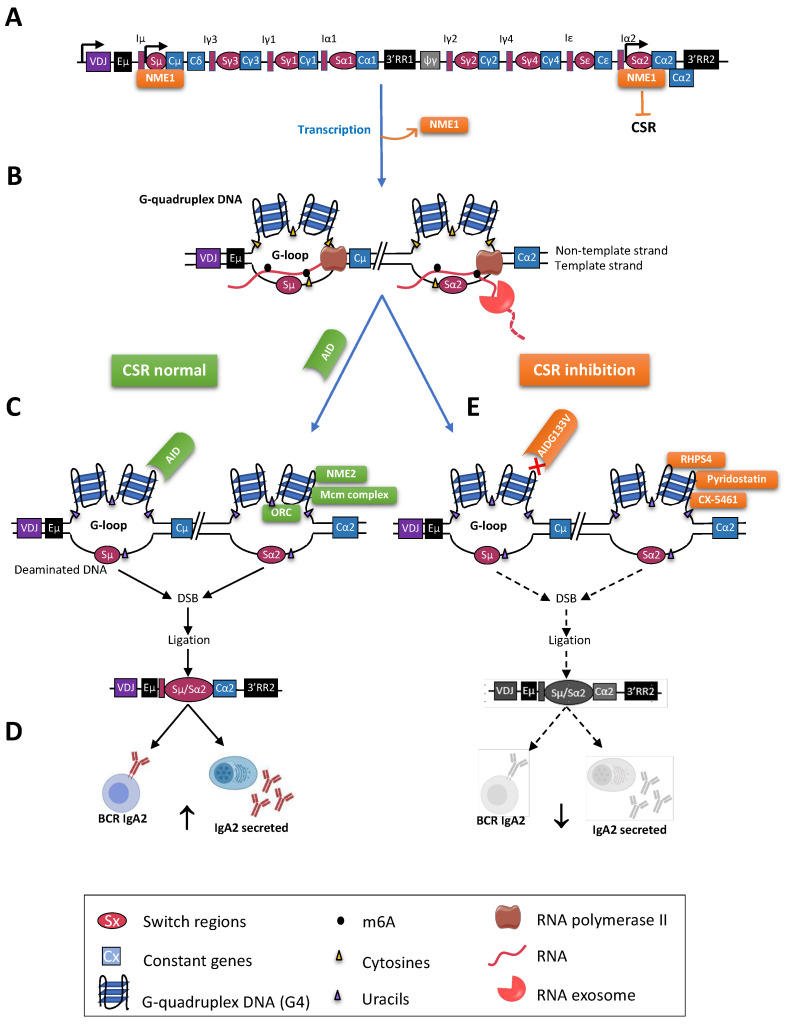
G4s and class switch recombination. (**A**) The human IgH locus is represented after VDJ recombination. The recombined VDJ gene and C_H_ genes are represented by rectangles and the S regions by ovals. S regions are preceded by promoters and I exons. The human IgH locus also includes two regulatory 3′RR (black rectangles). Before CSR, NME1 binds to the S regions and prevents CSR; (**B**) Transcription through S regions by RNA Pol II (brown) yields noncoding RNA (purple). R-loops facilitate the formation of G4s (G-loops). Within loops, RNA:DNA hybrids restrict the accessibility of AID to only the nontemplate strand. The RNA exosome (orange) degrades the RNA hybridized to the template DNA strand, also exposing it to AID for DNA deamination; (**C**) After B cell activation, NME1 is removed, and AID binds G4s of targeted S regions, initiating breaks to be repaired by ligation of distant S regions (here, Sµ and Sα2). AID targets cytosines (**C**) on accessible ssDNA. CSR is modulated by natural G4s ligands, such as NME2, which bind to G4s after transcription and stimulation, and also by the ORC and the Mcm complexes; (**D**) CSR diversifies the functions of B cells and of class-switched antibodies (here IgA2); (**E**) CSR can be inhibited by chemicals ligands of G4s, such as RHPS4, pyridostatin, and CX-5461, decreasing the frequency of B cells expressing or secreting class-switched Ig.

## Data Availability

Not applicable.

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
