# Peer review of "Roles of G4-DNA and G4-RNA in Class Switch Recombination and Additional Regulations in B-Lymphocytes"

_molecules, 2023, doi:10.3390/molecules28031159_

Round 1

Reviewer 1 Report

This is a very interesting review on a relevant subject that has not received a lot of attention until thus far, and should therefore be published. The review provides a comprehensive overview of the current knowledge on the role of G4-structured nucleic acid in class switch recombination and related DNA recombination mechanisms in B cells.

I have a few minor remarks and suggestions to improve the flow and readability of this review.

1. The subject encompasses the role of G4-structured DNA and RNA in B cells, whereas in the title only G4-DNA is mentioned, perhaps the title could be adjusted to include G4-RNA?

2. In section 1 it is mentioned that some of the general roles of G4 are beyond the scope of this review, yet in section 2 some of these general features are extensively elaborated upon. This section could be condensed somewhat further to create a better focus on the role of G4 in B cells.

3. A more thorough and legible introduction of the class switch recombination mechanism in section 3 would be helpful. Also, in section 3.1 somatic hypermutation appears out of the blue, this requires some more introduction. In general, this section needs to better structured.

4. The transcriptional regulation of CSR in the context of locus accessibility receives a lot of attention, while it is not made sufficiently clear how this connects to G4 DNA, and what the specific role of G4 DNA in this process is. This section appears somewhat lopsided as it quite lengthy in comparison to other aspects involved in CSR that are discussed. 

5. Section 4 provides the primary rationale for this review, for a more logical flow it could be considered to move section 4.1 up and incorporate it earlier in this review.

6. I find figure 1 quite general and not very informative, it is unclear whether (and how) the different biological functions are regulated by G4-DNA, G4-RNA or hybrids. Please consider revising this figure to convey a more specific message.

7. Line 35: define 'regulatory sequences'.

8. Line 40: omit the word 'physiological'.

9. Line 72: exchange 'strings' for 'runs'.

10. Line 74: please define and elaborate on 'G4-stabilizing ligands'.

11. Line 117: please omit the word 'will'.

12. Line 136 please exchange 'hypoxia' for 'hypoxic'.

13. Line 151: please exchange 'finely' for 'tightly'.

14. Lines 157-166: this part is repetitious, please revise.

15. Lines 189-190: this part is unclear, what is the point to be made here?

16. Line 213: What is meant by 'the addition of Salpha transcripts in trans'? Please elaborate on the experiment that is discussed here to provide clarity.

17. Paragraph 3.3 (lines 220-243): the link between the 3'RR and G4s is not made sufficiently clear from in this paragraph. How does it relate to G4-structured nucleic acid?

18. Positive and negative regulatory elements cannot act synergistically, perhaps the phrase antagonistically is meant? Please revise.

19. Lines 306-307: Please elaborate on the 'dedicated mouse models', this statement is too vague as is.

20. Line 340: Please explain (or replace) the term 'paradigmatic model', what is meant by this?

21. Line 451: please exchange 'is' for 'in'.

22. Line 468: please exchange 'amount' for 'stretch'.

Author Response

We thank the reviewer for his/her valuable comments and we revised our manuscript accordingly. Please find our answers in the attached document.

Reviewer 2 Report

Thank you for the opportunity to review the manuscript entitled ‘G4-DNA, class switch recombination and potential additional regulations in B-lymphocytes’ by Deze et al. The manuscript does indeed review specifically the issue mentioned in the title. It provides perspective and much of current understanding of the biological roles these architectures are known for.

 Its main weaknesses are its introduction and conclusions. Indeed, some mistakes appear in the introductory paragraphs. In the first paragraph, the manuscript starts with an incorrect definition of G-quadruplexes. As the authors state in pg 3, line 70, G-quadruplexes may form with N = 2. Therefore the citation is also incorrect- or at a minimum incomplete. This definition also does not include hybrid RNA:DNA, or more generally those architectures formed by more than one polymeric sequence. Furthermore, it is simply not correct to cite yet-to-be-demonstrated functional role for quadruplexes of (G:C:G:C) tetrads to a publication dated from 2001. Another of these mistakes- first sentence of second paragraph features an incorrect citation once again; second sentence missing citations.

And finally, the perspective taken by the authors in the Conclusions addresses G4 as druggable architectures. Regrettably, since selectivity is not discussed in the manuscript this approach diminishes the contribution of the paper to the greater understanding it provides.

 Reference 58 is repeated (see 62)

Author Response

We thank the reviewer for his/her positive appreciation and we have modified the manuscript according to the pertinent and helpful comments that were made.
